# Artificial Selection on Microbiomes To Breed Microbiomes That Confer Salt Tolerance to Plants

Ulrich G. Mueller,[a] Thomas E. Juenger,[a] Melissa R. Kardish,[a,b] Alexis L. Carlson,[a] Kathleen M. Burns,[a] Joseph A. Edwards,[a] Chad C. Smith,[a] Chi-Chun Fang,[a] David L. Des Marais[a,c]

[a]Department of Integrative Biology, University of Texas at Austin, Austin, Texas, USA
[b]Center for Population Biology, University of California, Davis, California, USA
[c]Department of Civil & Environmental Engineering, Massachusetts Institute of Technology, Cambridge, Massachusetts, USA

**ABSTRACT** We develop a method to artificially select for rhizosphere microbiomes that confer salt tolerance to the model grass *Brachypodium distachyon* grown under sodium salt stress or aluminum salt stress. In a controlled greenhouse environment, we differentially propagated rhizosphere microbiomes between plants of a nonevolving, highly inbred plant population; therefore, only microbiomes evolved in our experiment, but the plants did not evolve in parallel. To maximize microbiome perpetuation when transplanting microbiomes between plants and, thus, maximize response to microbiome selection, we improved earlier methods by (i) controlling microbiome assembly when inoculating seeds at the beginning of each selection cycle; (ii) fractionating microbiomes before transfer between plants to harvest, perpetuate, and select on only bacterial and viral microbiome components; (iii) ramping of salt stress gradually from minor to extreme salt stress with each selection cycle to minimize the chance of overstressing plants; (iv) using two nonselection control treatments (e.g., nonselection microbial enrichment and null inoculation) that permit comparison to the improving fitness benefits that selected microbiomes impart on plants. Unlike previous methods, our selection protocol generated microbiomes that enhance plant fitness after only 1 to 3 rounds of microbiome selection. After nine rounds of microbiome selection, the effect of microbiomes selected to confer tolerance to aluminum salt stress was nonspecific (these artificially selected microbiomes equally ameliorate sodium and aluminum salt stresses), but the effect of microbiomes selected to confer tolerance to sodium salt stress was specific (these artificially selected microbiomes do not confer tolerance to aluminum salt stress). Plants with artificially selected microbiomes had 55 to 205% greater seed production than plants with unselected control microbiomes.

**IMPORTANCE** We developed an experimental protocol that improves earlier methods of artificial selection on microbiomes and then tested the efficacy of our protocol to breed root-associated bacterial microbiomes that confer salt tolerance to a plant. Salt stress limits growth and seed production of crop plants, and artificially selected microbiomes conferring salt tolerance may ultimately help improve agricultural productivity. Unlike previous experiments of microbiome selection, our selection protocol generated microbiomes that enhance plant productivity after only 1 to 3 rounds of artificial selection on root-associated microbiomes, increasing seed production under extreme salt stress by 55 to 205% after nine rounds of microbiome selection. Although we artificially selected microbiomes under controlled greenhouse conditions that differ from outdoor conditions, increasing seed production by 55 to 205% under extreme salt stress is a remarkable enhancement of plant productivity compared to traditional plant breeding. We describe a series of additional experimental protocols that will advance insights into key parameters that determine efficacy and response to microbiome selection.

Address correspondence to Ulrich G. Mueller, umueller@austin.utexas.edu.

**KEYWORDS** beneficial microbes, *Brachypodium distachyon*, experimental evolution, host-mediated indirect selection, microbiome selection, rhizosphere microbiome, salt stress, salt tolerance, microbiome breeding

A challenge in plant-microbiome research is engineering of microbiomes with specific and lasting beneficial effects on plants. These difficulties of microbiome engineering derive from several interrelated factors, including transitions in microbiome function during plant ontogeny and the complexity of microbiome communities, such as hyperdiverse rhizosphere or phyllosphere microbiomes containing countless fungal, bacterial, and viral components (1–3). Even when beneficial microbiomes can be assembled experimentally to generate specific microbiome functions that benefit a plant, microbiomes are often ecologically unstable and undergo turnover (i.e., microbiome communities change dynamically over time), for example, when new microbes immigrate into microbiomes, when beneficial microbes are lost from microbiomes, or when beneficial microbes evolve new properties under microbe-microbe competition that are detrimental to a host plant.

One strategy to engineer sustainable beneficial microbiome function uses repeated cycles of differential microbiome propagation to perpetuate between hosts only those microbiomes that have the most desired fitness effects on a host (Fig. 1). Such differential propagation of microbiomes between hosts can therefore artificially select for microbiome components that best mediate stresses that impact host fitness (4–7). Only three experimental studies have used this approach so far for plants. Two studies selected on rhizosphere microbiomes of the plant *Arabidopsis thaliana* (4, 8), and both studies needed more than 10 cycles of microbiome selection to generate a modest and highly variable phenotypic response in plant phenotypes (e.g., increase in aboveground biomass by ~10%) (4). A third study (9) used seven cycles of microbiome selection to generate microbiomes that significantly delayed the onset of drought symptoms of water-stressed wheat plants. Here, we expand on these studies to artificially select for bacterial rhizosphere microbiomes that confer salt tolerance to the model grass *Brachypodium distachyon* (Fig. 1). Our methods specifically aim to improve microbiome perpetuation between plants and to optimize response to artificial microbiome selection by controlling microbiome assembly when inoculating seeds, using low-carbon soil to enhance host control exerted by seedlings during initial microbiome assembly and early plant growth, harvesting and perpetuating microbiomes that are in close physical contact with plants, short cycling of microbiome generations to select for microbiomes that benefit seedling growth, and ramping of salt stress between selection cycles to minimize the chance of either understressing or overstressing plants.

To optimize microbiome selection experiments, we found it useful to conceptualize the process of microbiome selection within a host-focused quantitative genetic framework (6) rather than within a multilevel selection framework preferred by Swenson et al. (4) (artificial ecosystem selection; see also reference 10). Both frameworks capture the same processes (i.e., neither framework is wrong), but a host-focused quantitative genetic framework is more useful to identify factors that can be manipulated to increase efficacy of microbiome selection. First, because microbiome selection aims to shape a fitness component of the host plant (e.g., stress tolerance) and because it is typically easier to measure plant phenotypes rather than measure microbiome properties, selection is indirect. Microbiomes are not measured directly, but microbiomes are evaluated indirectly by measuring host performance. Indirect selection is an established breeding technique that can be used when the target trait is difficult or costly to measure (11), as is the case for microbiome traits compared to the ease of measuring a host phenotype that is dependent on microbiome properties. The efficacy of indirect selection depends on strong correlations between microbiome and host traits; therefore, indirect microbiome selection should be more efficient if such correlations can be maximized experimentally, for example, by controlling ecological priority effects during initial microbiome assembly (12–15) or by increasing host control over microbiome assembly and persistence (14, 16). Second, because a typical host likely experienced a long history of evolution to monitor and manipulate its microbiomes (a process called host control) (16–19), indirect microbiome

mSystems®

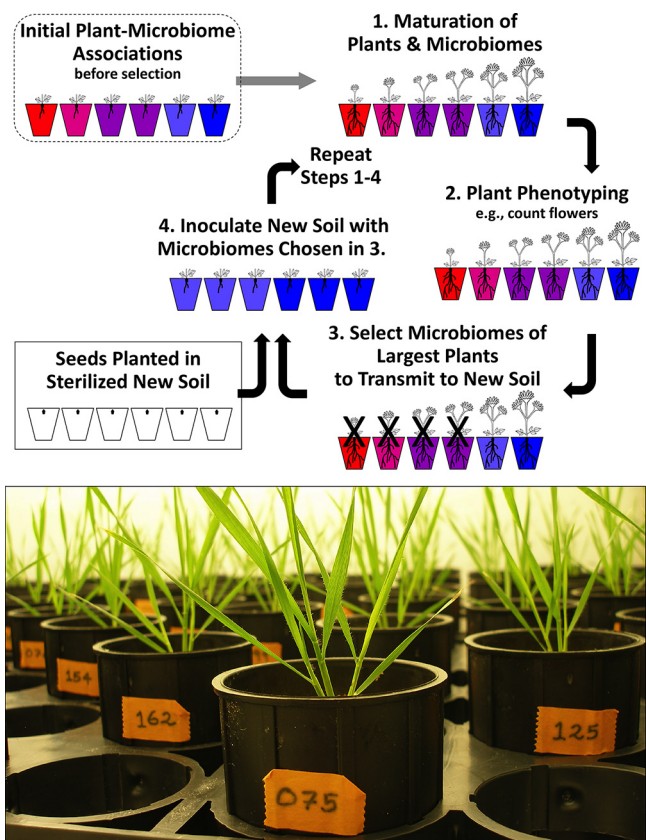

**FIG 1** Host-mediated artificial selection on microbiomes. (Top) Method of differential microbiome propagation to impose artificial selection on rhizosphere microbiomes (modified from reference 6 with permission of the publisher). The host plant does not evolve because this method harvests microbiomes from mature plants and propagates these microbiomes to sterilized seeds planted in sterilized soil (step 4), but seeds are taken each cycle from a nonevolving source (stored seeds). The method imposes indirect selection on microbiomes because microbiome properties are not measured directly; instead, microbiome effects are estimated indirectly by measuring host fitness (e.g., plant biomass); therefore, host fitness is used as an indicator to infer association with rhizosphere microbiomes that benefit a plant. Both evolutionary and ecological processes can alter microbiomes at each step in the cycle (see the text), but at steps 3 and 4 in each cycle, experimental protocols aim to maximize evolutionary changes stemming from differential microbiome propagation. (Bottom) Experimental plants of the model grass *Brachypodium distachyon* shortly before harvesting of rhizosphere microbiomes for differential microbiome propagation. Photo by U.G.M.

selection uses the host as a kind of thermostat to help gauge and adjust the temperature of its microbiomes, and then propagate desired microbiomes between hosts (Fig. 1). Based on previous theories (5, 6, 20), such host-mediated indirect selection on microbiomes can be easier than direct selection on microbiomes, particularly with host species that exert strong host control over assembly and stability of their microbiomes (6, 13, 14, 21).

Microbiome engineering by means of differential microbiome propagation (Fig. 1) alters microbiomes through both ecological and evolutionary processes. Ecological processes include changes in community diversity, relative species abundances, or structure of microbe-microbe or microbe-plant interaction networks. Evolutionary processes include extinction of specific microbiome members; allele frequency changes, mutation, or gene transfer between microbes; and differential persistence of microbiome components when differentially propagating microbiomes at each selection cycle. These processes can be interdependent (e.g., in the case of ecoevolutionary feedback [22, 23]), and some processes can be called either ecological or evolutionary (e.g., loss of a microbe from a microbiome can be viewed as evolutionary extinction or as an outcome of ecological competition), but for the design of a microbiome selection protocol, it is useful to think about ecological processes separately from evolutionary processes. Microbiome selection protocols aim to maximize changes in the genetic makeup of microbiomes through differential microbiome propagation (steps 3

and 4 in Fig. 1), for example, by optimizing microbiome transmission during microbiome transplanting between hosts or by optimizing microbiome reassembly after such transfers (e.g., by facilitating ecological priority effects at host inoculation). Although both evolutionary and ecological processes alter genetic makeup of microbiomes during each propagation cycle (Fig. 1), as shorthand, we refer to the changes resulting from host-mediated indirect selection on microbiomes as microbiome response due to microbiome selection.

## RESULTS

**Artificially selected microbiomes confer increased salt tolerance to plants.** Figure 2 shows the changes in relative plant fitness (aboveground dry biomass) during eight rounds of differential microbiome propagation. Relative to fallow-soil control (nonselection enrichment) treatment and null control treatment, selected microbiomes confer increased salt tolerance to plants after only 1 to 3 selection cycles for both the sodium stress (Fig. 2a and c) and the aluminum stress treatments (Fig. 2b and d). Relative to fallow-soil control plants, artificially selected microbiomes increase plant fitness by 75% under sodium sulfate stress ($P < 0.001$) and by 38% under aluminum sulfate stress ($P < 0.001$). Relative to null control plants, selected microbiomes increase plant fitness by 13% under sodium sulfate stress and by 12% under aluminum sulfate stress. Although repeated rounds of differential microbiome propagation improved plant fitness between successive microbiome generations (particularly relative to the null controls; Fig. 2c and d), interactions between treatment and generation were not statistically significant (see Text S3 in the supplemental material). This implies that fitness-enhancing effects of microbiomes from selection lines were realized after one or a few rounds of microbiome selection (e.g., Fig. 2c and d), and there was insufficient statistical support that, under the gradually increasing salt stress, any additional rounds further resulted in greater plant biomass of selection lines relative to control lines. However, because plants were exposed to increasingly greater salt stresses in later generations (Fig. 2e and f, Text S1), selected microbiomes of later generations helped plants tolerate more extreme salt stresses.

The phenotypic effect on plants due to the evolving microbiomes fluctuated during the eight rounds of differential microbiome propagation (Fig. 2a to d). Such fluctuations can occur in typical artificial selection experiments (24), but fluctuations may be more pronounced when artificially selecting on microbiomes (25) because additional factors can contribute to between-generation fluctuations. Specifically, across the eight selection cycles in our experiment, the observed fluctuations could have been due to (i) uncontrolled humidity changes and correlated humidity-dependent water needs of plants (humidity was not controlled in our growth chamber), consequently changing the effective salt stresses; (ii) the strong ramping of salt stress during the first five selection cycles, possibly resulting in excessively stressed plants in generations 4 and 5 (see discussion in Text S1); (iii) random microbiome changes (microbiome drift) and consequent random microbe-microbe interactions; or (iv) other such uncontrolled factors. The fluctuations in plant fitness are most prominent during the first five selection cycles (Fig. 2a to d) when we increased salt stress 2- to 5-fold between generations and when humidity varied most in our growth chamber (Text S1), whereas fluctuations were less pronounced during the last three generations when we changed salt stress only minimally and humidity was relatively stable. These observations are consistent with known responses of B. distachyon to environmental stresses (26), predicting that artificial selection on microbiomes conferring salt tolerance to plants should be most efficient under experimental conditions that rigorously control soil moisture, salt stress, humidity, and plant transpiration.

**Effect of artificially selected microbiomes on seed production.** In the last microbiome generation after a ninth microbiome selection cycle (generation 9), we grew plants for 68 days to quantify the effect of our artificially selected microbiomes on seed production. We also added one control treatment, solute transfer control (solute control), to help elucidate some of the mechanisms underlying the salt tolerance-conferring effects of selected microbiomes on seed production (Fig. 3). In solute control

## Sodium-Salt Tolerance

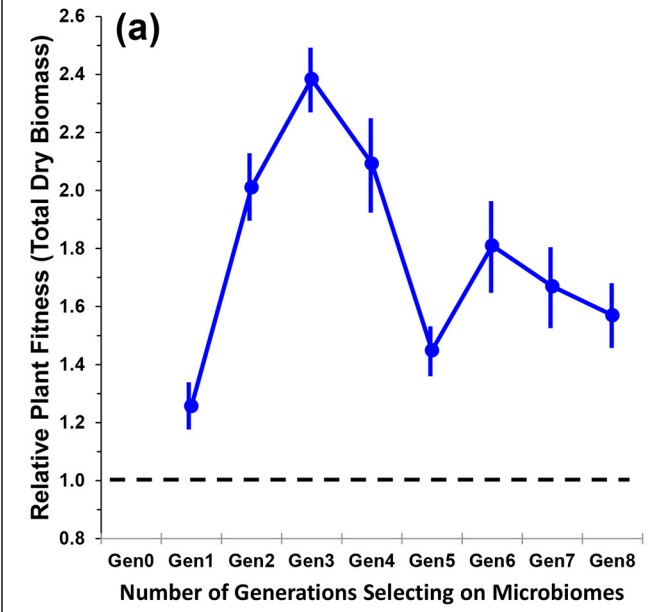

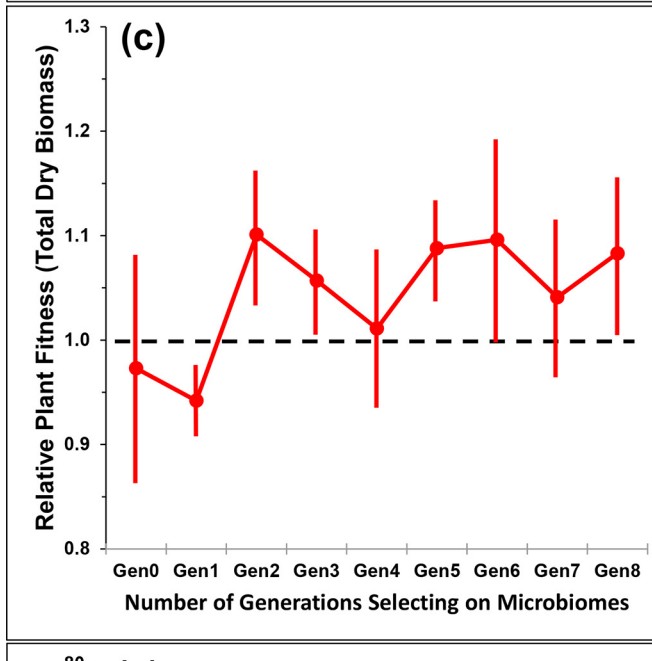

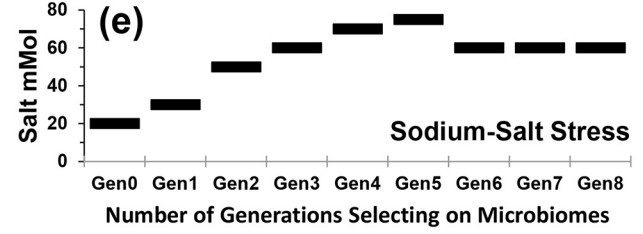

## Aluminum-Salt Tolerance

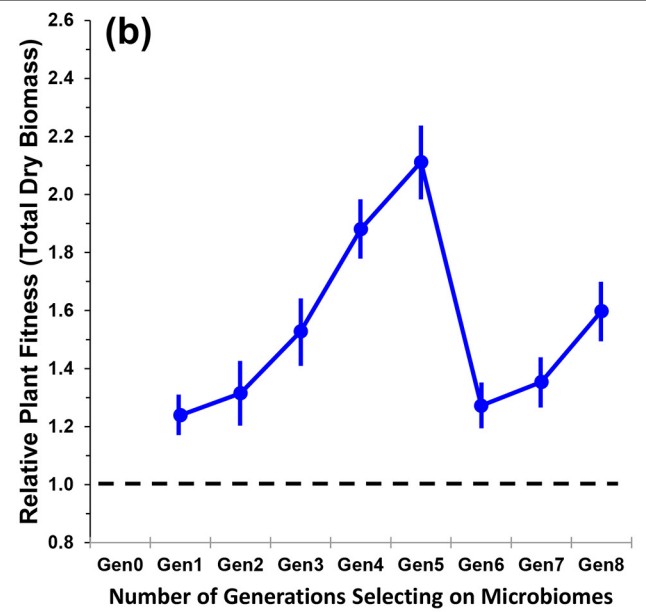

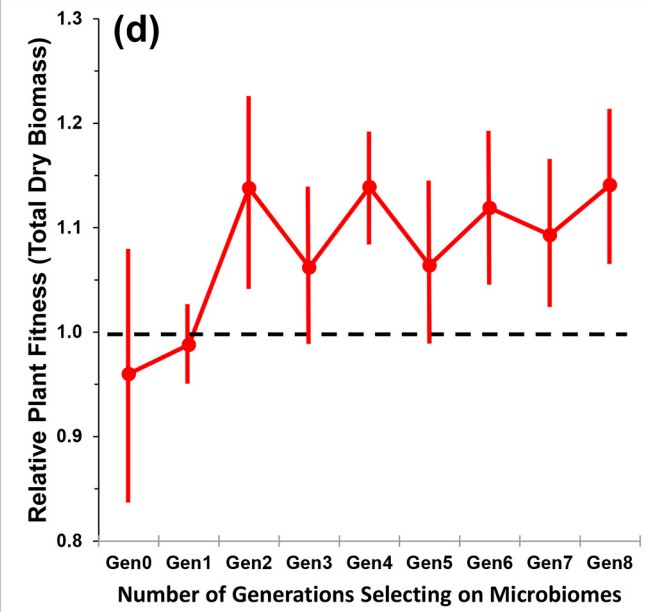

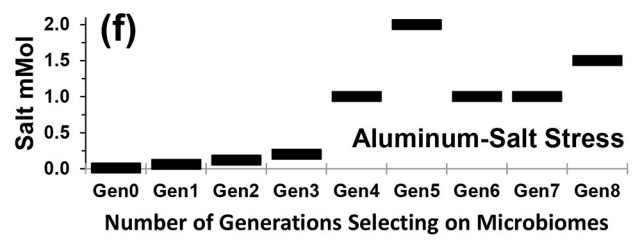

**FIG 2** Artificial selection on microbiomes to generate microbiomes that confer salt tolerance to plants. Microbiomes were artificially selected in two concurrent experiments under either sodium salt stress (left column) or aluminum salt stress (right column). After microbiome inoculation of plants in the baseline generation (Gen0), microbiomes were propagated differentially for 8 selection cycles (generations, Gen), using the microbiome propagation scheme in Fig. 1. Two salt stresses, sodium sulfate stress (a, c, and e) and aluminum sulfate stress (b, d, and f), were imposed in parallel in different lines of microbiome selection. Fitness of plants receiving artificially selected microbiomes is shown in panels a to d relative to two nonselected control treatments. In fallow-soil microbiome propagation control, microbiomes were harvested from fallow soil (soil in pot with no plant) and then propagated to sterile

treatments, we eliminated with 0.2-$\mu$m filters live cells from the harvested micro-biomes in the selection lines to test the growth-enhancing effects of root exudates and viruses that may be copropagated with bacterial microbiomes in the selection lines. Plants receiving these bacterium-free, filtered solutes had (i) significantly poorer seed production than plants that received these same solutes together with the live bacterial microbiomes ($P < 0.02$ for sodium stress treatment; $P < 0.05$ for aluminum stress treatment; Text S3) and (ii) seed production that was comparable to that of plants from null control treatments ($P > 0.7$ for sodium stress treatment; $P > 0.25$ for aluminum stress treatment; Text S3). These findings indicate that no plant exudates or viruses copropagated with bacterial microbiomes accounted for the salt tolerance-con-ferring effects of selected microbiomes and that any cotransplanted solutes (e.g., root exudates) and any copropagated viruses affected plant growth like the null control treatments (i.e., no exudates, no viruses).

**Specificity test by crossing evolved SOD and ALU microbiomes with SOD and ALU stress.** In the cross-fostering control of the last microbiome generation, we crossed harvested microbiomes from the sodium stress (SOD) and aluminum stress (ALU) selection lines with the two types of salt stress in soil to test specificity of the salt-ameliorating effects of the microbiomes (Fig. 4, Table S2). The effect of microbiomes selected to confer tolerance to aluminum sulfate appears nonspecific (aluminum-selected microbiomes appear to confer equal tolerance to both sodium and aluminum sulfate stress; $P > 0.5$; Fig. 4), but the effect of bacterial microbiomes selected to confer tolerance to sodium sulfate appears specific (sodium-selected microbiomes confer less tolerance to aluminum sulfate stress; $P < 0.002$; Fig. 4).

## DISCUSSION

Our study aimed to improve the differential microbiome propagation scheme that was originally developed by Swenson et al. (4) and then test the utility of our improved methods by artificially selecting on microbiomes to confer salt stress tolerance to plants. Swenson et al.'s original whole-soil community propagation scheme failed to generate consistent benefits for plant growth, and growth enhancement due to putatively selected communities was overall minor when averaged across all propagation cycles (average of ~10% growth enhancement). To address these problems, we adopted in our experiment ideas from quantitative genetics, microbial ecology, and host-microbiome evolution to optimize steps in our microbiome propagation protocol (Fig. 1), with the aim to improve perpetuation of beneficial microbiomes. Specifically, our methods aimed to (i) facilitate ec-ological priority effects during initial microbiome assembly (13, 14, 21), increasing micro-biome inheritance by steering the initial recruitment of symbiotic bacteria into rhizosphere microbiomes of seedlings; (ii) propagate microbiomes harvested from within the sphere of host control (i.e., microbiomes in close physical proximity to roots), whereas Swenson et al. (4) and Panke-Buisse et al. (8) harvested microbes from outside the sphere of host con-trol; (iii) enhance carbon-dependent host control of microbiome assembly and of micro-biome persistence by using low-carbon soil (1, 6, 27, 28); and (iv) gradually increase salt stress between selection cycles to minimize the chance of either understressing or over-stressing plant. Without additional experiments, it is not possible to say which of these ex-perimental steps was most important to increase response to microbiome selection. Because Jochum et al. (9) succeeded at artificially selecting for microbiomes that confer

**FIG 2** Legend (Continued)

fallow soil of the next microbiome generation. In the null control, plants did not receive microbiome inocula, but microbes could "rain in" from air, as in all treatments. Horizontal dashed lines in panels a to d indicate the threshold above which plants given selected microbiomes had higher relative fitness than control plants relative to fallow-soil control plants (a and b) and relative to null control plants (c and d). Each selection treatment had 5 selection lines (8 plants/line), and the error bars show the standard deviation from the 5 averages of these 5 selection lines. (e and f) Salt stresses were increased between selection cycles, starting with minor salt stresses, increasing gradually to minimize the chance of overstressing the plants but decreasing salt stress if plants seemed overstressed (details in the supplemental material). Because of the increasing salt stresses (e and f), selected microbiomes enabled plants to cope with more severe stresses and, therefore, had stronger fitness-enhancing effects on plants in later generations. Relative to fallow-soil control treatments, selected microbiomes increase plant fitness by 75% under sodium sulfate stress (a) and by 38% under aluminum sulfate stress (b). Relative to null control treatments, selected microbiomes increase plant fitness by 13% under sodium sulfate stress (c) and by 12% under aluminum sulfate stress (d).

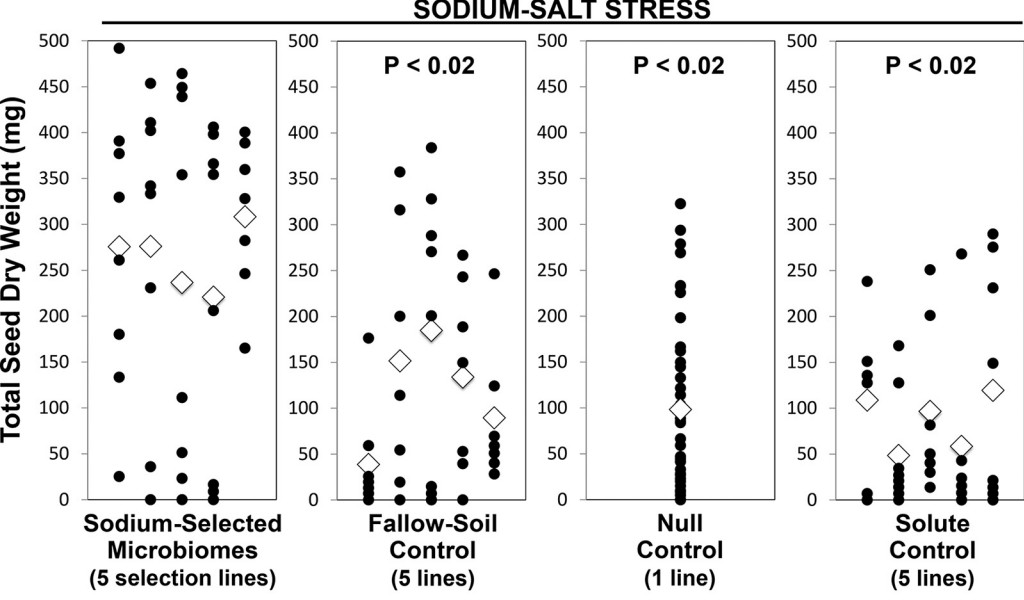

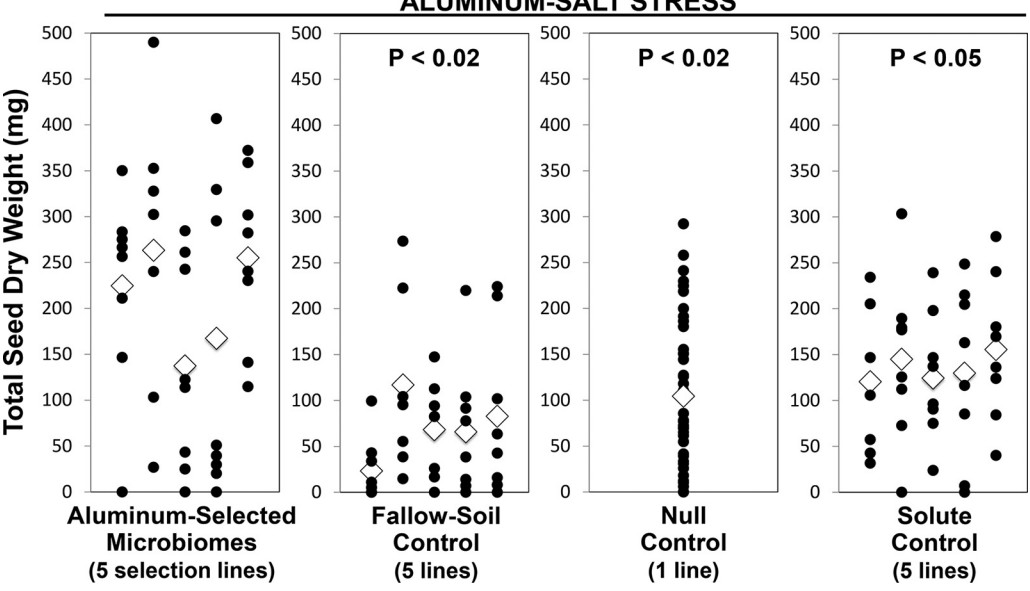

**FIG 3** Artificially selected microbiomes increase seed production under salt stress. At the end of our experiment after a ninth selection cycle (generation 9), plants were grown to seed for 68 days to test whether rhizosphere microbiomes selected to increase aboveground biomass of preflowering plants generated microbiomes that also enhance seed production. Total seed dry weight is plotted as a black dot for each plant; plants of the same selection line are plotted vertically above each other; and the average for each line is plotted as a diamond. Overlapping data points are adjusted here minimally to separate such data points and visualize all data points. In addition to fallow-soil control and null control used in generations 1 to 8 (Fig. 2), solute control was added in generation 9. In solute control, selected bacterial microbiomes harvested from rhizospheres were filtered to remove all bacterial components to test for any growth-enhancing effects of viruses and solutes (e.g., plant hormones exuded into soil) that are unavoidably copropagated with any harvested rhizosphere microbiome. All controls are significantly different from the corresponding selection treatment (leftmost panel); $P$ values are shown above each control, and $P$ values are corrected using the false discovery rate for *post hoc* comparisons (Text S3). Plants were salt stressed because many plants never produced seeds (or few seeds; see also Fig. S5, top left), whereas essentially all plants would produce many seeds under stress-free conditions. Artificially selected microbiomes helped plants cope with these salt stresses, because plants that received selected microbiomes outperformed plants of all three control treatments, including solute control plants (indicating that selected bacterial microbiomes conferred salt tolerance to plants rather than any copropagated viruses). Seed production of solute control plants is indistinguishable from the corresponding null control plants ($P = 0.71$, sodium salt stress; $P = 0.29$, aluminum salt stress; Text S3), indicating that plants receiving bacterium-free filtrate performed as if they had received a null control treatment. Although microbiomes were selected to increase aboveground biomass of preflowering plants (20 to 30 days old), selected microbiomes also enhanced seed production of older plants (68 days old).

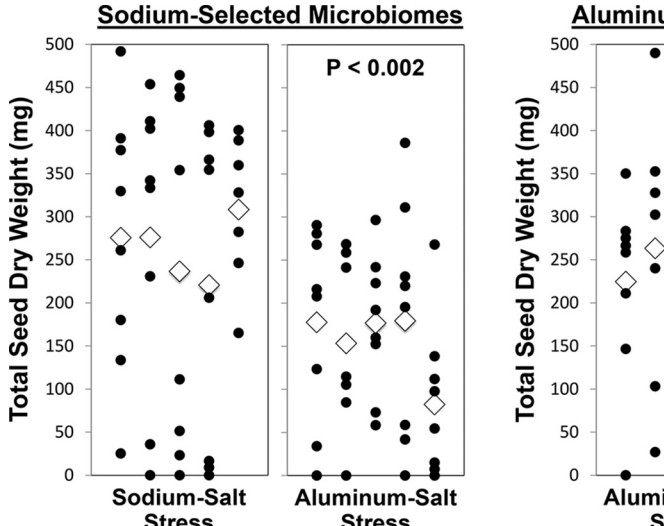
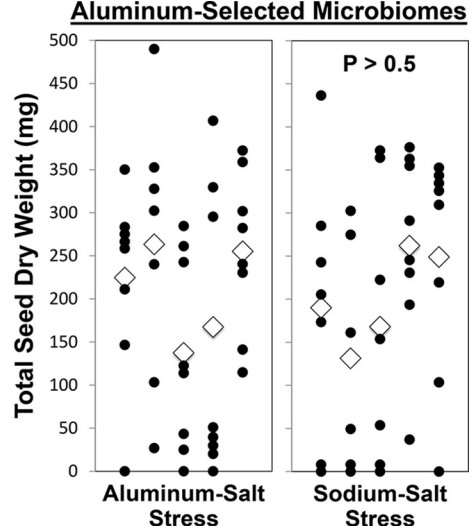

**FIG 4** Specific and nonspecific growth-enhancing effects of artificially selected microbiomes. In generation 9, a 2 by 2 cross-fostering experiment tested whether microbiomes selected under sodium salt stress conferred greater salt tolerance to plants stressed with sodium salt compared to plants stressed with aluminum salt and, conversely, whether microbiomes selected under aluminum salt stress conferred greater salt tolerance to plants stressed with aluminum salt compared to plants stressed with sodium salt. $P$ values are shown for each comparison, and $P$ values are corrected using the false discovery rate for *post hoc* comparisons (Text S3). The effect of microbiomes selected to confer tolerance to aluminum salt appears to be nonspecific because these microbiomes confer equal tolerance to plants stressed with either sodium salt or aluminum salt ($P > 0.5$; two rightmost panels), whereas the effect of bacterial microbiomes selected to confer tolerance to sodium salt appears specific, because these sodium-selected microbiomes confer less salt tolerance, or confer no salt tolerance, to plants under aluminum salt stress ($P < 0.002$; two leftmost panels).

drought tolerance to wheat grown in high-carbon soil, either low-carbon soil may not be essential for plant-mediated microbiome selection, contrary to our assumption, or high-carbon soil may facilitate microbiome selection of fungal components, because Jochum et al. (9) propagated between generations both bacterial and eukaryote rhizosphere components.

Compared to two earlier experiments of host-mediated microbiome selection by Swenson et al. (4) and Panke-Buisse et al. (8), our selection scheme appears to generate more pronounced and more stable effects on plant phenotype as a result of host-mediated microbiome selection. Except for the initial two selection cycles (Fig. 2a to d), our selected microbiomes consistently outperformed in subsequent selection cycles of the nonselected microbiomes of the control conditions. In contrast, for example, Swenson et al.'s (4) experiments sometimes resulted in selected microbiomes that were outperformed by control microbiomes. Our methods may have generated more stable microbiome effects because (i) only bacteria but no fungi were propagated between generations (Swenson et al. suspected fungal disease as a cause of occasional devastation of plant populations); (ii) we conducted our experiment in a more stable growth environment; and (iii) we selected for microbiomes conferring specific benefits (salt tolerance) rather than the nonspecific, general-purpose beneficial microbiomes selected by Swenson et al. (4) and Panke-Buisse et al. (8). After only 1 to 3 selection cycles, our selected microbiomes consistently outperformed the control microbiomes, with averages of 75% (SOD) and 38% (ALU) growth improvement relative to fallow-soil controls and 13% (SOD) and 12% (ALU) growth improvement relative to null controls (Fig. 2a to d). Most importantly, when quantifying plant fitness by total seed production in the final generation 9, plants with selected microbiomes outperformed fallow-soil controls, null controls, and solute controls by 120 to 205% (SOD) and 55 to 195% (ALU) (Fig. 3). Although we achieved these results under controlled greenhouse conditions that are very different from outdoor conditions, this seems a remarkable enhancement of plant productivity compared to traditional plant breeding.

An interesting result is that microbiomes selected to benefit growth of plants during the early vegetative phase (biomass of ~4-week-old plants, well before flowering; Fig. 1)

generated microbiomes that enhanced plant fitness during the reproductive phase by increasing the seed set of 10-week-old plants (Fig. 3). Rhizosphere microbiomes of grasses can change significantly during plant ontogeny (29); therefore, microbiomes selected to serve one function, such as early growth, may not necessarily optimize other functions, such as seed set. Therefore, the finding that microbiome selection to promote early growth (Fig. 2) also promotes increased seed set (Fig. 3) implies that (i) seed set is intrinsically tied to optimal early growth in *B. distachyon*, possibly by accelerating the timing of flowering; (ii) some of the same bacteria benefitting plants during the early vegetative phase also benefit plants during the reproductive phase, despite overall microbiome changes during plant ontogeny; and (iii) microbiome selection experiments aiming to increase seed productivity do not necessarily have to select on seed set as a measured phenotype but can shorten each selection cycle by selecting other phenotypes measurable during early vegetative growth.

Because Jochum et al.'s (9) and our experiments were the first systematic attempts to improve the methods of Swenson et al. (4), we predict that it should be possible to further optimize protocols of differential microbiome propagation. Microbiome selection therefore could emerge as a novel tool to elucidate microbiome functions in controlled laboratory environments and possibly also in those natural environments that allow control of key parameters affecting microbiome harvest, microbiome transfer, and microbiome inheritance. Such optimization of microbiome selection should ideally be informed by metagenomic analyses of experimental contrasts (e.g., comparison of microbiomes selected to confer tolerance to either sodium stress or aluminum stress) and by time-series analyses across microbiome propagation cycles to identify candidate microbes and microbial consortia important in mediating stresses.

**Additional experiments to improve methods of microbiome selection.** To expand on our methods of artificial microbiome selection, we outline here a series of additional experiments that should generate insights into key parameters that determine efficacy of microbiome selection. Arias-Sánchez et al. (7), Xie et al. (30), Chang et al. (31, 32), and Sánchez et al. (33) recently summarized criteria for microbiome selection experiments that are not host mediated (e.g., selection on $CO_2$ emission by a microbiome in the absence of a plant host); Lawson et al. (34) summarized protocols for engineering any kind of microbiome (e.g., using bottom-up and top-down design criteria); Henry et al. (35), Arora et al. (36), and Henry and Ayroles (37) developed methods for host-mediated microbiome selection using *Drosophila* as a host; and we focus below on methods of host-mediated microbiome selection to improve performance of a plant host. Because host-mediated microbiome selection leverages traits that evolved to recruit and control microbiomes (so-called host control [6, 19, 38]), the first four experiments outlined below explore whether factors promoting strong microbiome control by a plant host could improve efficacy of microbiome selection.

**(i) Artificial microbiome selection on endophytic versus rhizosphere microbiomes.** Microbiomes internal to a host (e.g., endophytic microbes of plants) require some form of host infection and, therefore, could be under greater host control than external microbiomes, such as rhizoplane or rhizosphere microbiomes. Consequently, under stresses that are mediated by host-controlled microbes, it may be easier to obtain a response to microbiome selection when targeting selection on endophytic microbiomes. This prediction can be tested in an experiment that compares, in separate selection lines, the responses to microbiome selection when harvesting and propagating only endophytic microbiomes versus only rhizosphere microbiomes. This prediction may not hold for stresses that require stress mediation by microbes in the external microbiome compartment of roots (e.g., microbes that detoxify toxins, such as aluminum, before they enter the root and then affect the plant negatively, for example, microbes that chelate toxins external to the plant in the rhizosphere [39]); however, this prediction about a key role of host control for the efficacy of microbiome selection should hold for many other stresses that are mediated by microbes that a plant permits to enter into the endophytic compartment.

**(ii) Microbiome selection in two genetic backgrounds differing in host control.** A second approach to test for the role of host control is to compare microbiome selection in two different host genotypes, such as two inbred strains of the same plant species. For example, different host genotypes may recruit different kinds of microbes into symbiosis

(40). Such differences in host-controlled microbiome recruitment could result in differences in microbiome selection, and a microbiome artificially selected within one host genotype to improve one particular host trait may produce a different phenotypic effect when tested in a different host genotype.

(iii) **Varying host control by varying carbon content in soil.** A third approach to test host control is to compare the efficacy of microbiome selection in low- versus high-carbon soil. Microbial growth in some soils is limited by carbon, and many plants therefore regulate their soil microbiomes by carbon exudates (41). We therefore hypothesized that a low-carbon soil (like the carbon-free soil in our experiment) facilitates host control and consequently also microbiome selection. This hypothesis remains to be tested in, for example, a microbiome selection experiment contrasting response to selection when using soils with different carbon contents. Because Jochum et al. (9) recently showed that it is possible to artificially select for microbiomes that confer drought tolerance to wheat grown in high-carbon soil, low-carbon soil may not be essential for plant-mediated microbiome selection, but low-carbon soil could be a facilitating condition.

(iv) **Manipulating resource-limited host control by varying seed size.** A fourth approach to test host control could be to compare the efficacy of microbiome selection between plant species with large seeds versus small seeds (e.g., *Brachypodium* versus *Arabidopsis*) or between seedlings of the same species grown from small versus large seeds. A germinating seed has to allocate resources to aboveground growth to fix carbon and to belowground growth to access nutrients and water, and seedlings growing from resource-rich large seeds therefore may be better able to allocate resources to manipulate microbiomes effectively, for example, by root exudates. If such resource allocation constraints exist for young seedlings, this could explain why our microbiome selection experiment with *B. distachyon* appears to have generated stronger and faster response to microbiome selection than other such experiments with *Arabidopsis thaliana* (4, 8).

(v) **Propagation of fractionated versus whole microbiomes.** Experimental microbiome propagation between host generations can be complete (all soil community members are propagated between hosts, as in Swenson et al. [4], Panke-Buisse et al. [8], and Jochum et al. [9]), or microbiomes can be fractionated by excluding specific microbial components, as in our protocol where we propagated only organisms of bacterial or smaller sizes. We used fractionated microbiome propagation because (i) we were more interested in elucidating contributions to host fitness of the understudied bacterial components than the fungal components (e.g., mycorrhizal fungi) and (ii) fractionation simplifies analyses of the microbiome responses to selection (e.g., bacterial microbiome components, but not necessarily fungal components, need to be analyzed with metagenomic techniques). However, because fungal components and possible synergistic fungal-bacterial interactions cannot be selected on when using our fractionated microbiome propagation scheme, we hypothesized previously (6) that selection on fractionated microbiomes shows attenuated selection responses compared to selection on whole microbiomes. This can be tested in an experiment comparing the response to microbiome selection when propagating fractionated versus whole microbiomes, for example, by using different size-selecting filters.

(vi) **Propagation of mixed versus unmixed microbiomes.** When propagating microbiomes to new hosts, it is possible to propagate mixed microbiomes harvested from different hosts or only unmixed microbiomes. Therefore, mixed versus unmixed propagation schemes represent two principal methods of microbiome selection (4–6, 42, 43). Compared to unmixed propagation, mixed propagation generated a faster response to microbiome selection for microbiomes propagated *in vitro* in the absence of a host (43), but the respective advantages of mixed versus unmixed propagation have yet to be tested for host-associated microbiomes, such as the rhizosphere microbiomes studied here. Mixed propagation may be superior to unmixed propagation if, for example, mixing generates novel combinations of microbes with novel beneficial effects on a host (6), may merge separate networks of microbes into a superior compound network (so-called community network coalescence; 42, 44), or may generate novel microbial interactions that increase microbiome stability (13).

(vii) **Microbiome diversity of the starter inoculum.** In our salt stress experiment, we aimed for a highly diverse starter microbiome to inoculate all pots of generation 0, but

we did not specifically try to include bacteria from sources that are most likely to include microbes that confer salt tolerance to plants. Could inclusion of microbiomes harvested from grasses growing naturally in salty soil have improved the diversity of bacteria in the starter inoculum and, thus, increased the response to microbiome selection in our experiment? Comparison of starter inocula harvested from plants growing naturally in salty versus nonsalty soils may be able to address this question.

## MATERIALS AND METHODS

We developed our microbiome selection protocol between 2011 and 2014 in a series of pilot experiments, conducted the microbiome selection experiment reported here between January and October 2015, and then disseminated our protocol via *bioRxiv* in 2016 (45) to facilitate teaching of workshops on microbiome selection. We describe here our experimental protocols, and a separate report (unpublished data) will describe the metagenomic analyses complementing the protocols and phenotypic results reported here.

**Maximizing microbiome perpetuation.** To select for microbiomes that confer salt tolerance to plants, we used a differential host-microbiome copropagation scheme as described in Swenson et al. (4), Mueller et al. (46), and Mueller and Sachs (6) but improved on these earlier selection schemes by (i) maximizing evolutionary microbiome changes stemming from differential propagation of whole microbiomes at step 3 in Fig. 1 while (ii) minimizing some, but not all, ecological microbiome changes that can occur at any of the steps in a selection cycle (e.g., we tried to minimize uncontrolled microbe-community turnover). In essence, our protocol aimed to maximize microbiome perpetuation (i.e., maximize inheritance of key microbes). To increase microbiome inheritance, we added protocol steps of known techniques, most importantly (i) facilitation of ecological priority effects during initial microbiome assembly (21), increasing microbiome inheritance by controlling in each selection cycle the initial recruitment of symbiotic bacteria into rhizosphere microbiomes of seedlings, and (ii) low-carbon soil to enhance carbon-dependent host control of microbiome assembly and microbiome persistence (1, 6, 27, 28). Theory predicts that any experimental steps increasing fidelity of microbiome perpetuation from mother microbiome to offspring microbiome should increase the efficacy of microbiome selection (6, 30, 35, 47).

**Maximizing microbiome heritability.** In each microbiome propagation cycle (microbiome generation), we inoculated surface-sterilized seeds taken from nonevolving stock (inbred strain Bd3-1 of the grass *Brachypodium distachyon*) (48), using rhizosphere bacteria harvested from roots of those plants within each selection line that exhibited the greatest aboveground biomass (Fig. 1). Microbiome selection within the genetic background of an invariant (i.e., highly inbred) plant genotype increases microbiome heritability, defined as the proportion of overall variation in the plant phenotype that can be attributed to differences in microbiome-encoded genetic effects on plants. By keeping plant genotype invariant, microbiome heritability increases because a greater proportion of the overall plant-phenotypic variation in a selection line can be attributed to differences in microbiomes. This increases an experimenter's ability to identify association with a desired microbiome (4), enhancing reliability of the plant phenotype as an indicator of microbiome effects and, thus, increasing efficacy of indirect selection on microbiomes.

**Harvesting rhizosphere microbiomes and selection scheme.** Each selection line consisted of a population of eight replicate plants, and each selection treatment had five replicate selection lines (i.e., 40 plants total per treatment). To determine phenotypes of plants on the day of microbiome harvesting, we judged aboveground growth visually by placing all eight plants of the same selection line in ascending order next to each other (see Fig. S3 in the supplemental material) and then choosing the two largest plants for microbiome harvest. For all plants, we cut plants at the soil level and then stored the aboveground portion in an envelope for drying and weighing. For each plant chosen for microbiome harvest, we extracted the entire root system from the soil and then harvested rhizosphere microbiomes immediately to minimize microbiome changes in the absence of a plant control. Root structures could be extracted whole because of a granular soil texture (profile porous ceramic soil), with some loss of fine roots. Because we were interested in harvesting microbiomes that were in close association with roots, we discarded any soil adhering loosely to roots, leaving a root system with few firmly attached soil particles. We combined the root systems from the two best-growing plants of the same selection line and harvested their mixed rhizosphere microbiomes by immersing and gently shaking the roots in the same salt nutrient buffer that we used to hydrate soils (details are in Text S1). Combining root systems from the two best-growing plants generated a so-called mixed microbiome harvested from two mother rhizospheres, which we then transferred within the same selection line to all eight offspring plants (i.e., germinating seeds) of the next microbiome generation (Fig. 1).

**Microbiome fractionation with size-selecting filters before microbiome propagation.** To simplify future metagenomic analyses from propagated microbiomes, we used 2-$\mu$m filters (details are in Text S1) to filter microbiomes harvested from rhizospheres of mother plants, thereby capturing only bacteria (and possibly also viruses) for microbiome propagation to the next microbiome generation but eliminating from propagation any larger-celled soil organisms (i.e., we excluded all eukaryote organisms in soil, including fungi). This fractionation step distinguishes our methods from those of Swenson et al. (4), Panke-Buisse et al. (8), and Jochum et al. (9), all of whom transferred between pots all organisms living in soil (including algae, nematodes, protozoans, fungi, etc.). Plant phenotypic changes in these previous experiments therefore were not necessarily due to changing microbiomes but possibly to eukaryotes that were copropagated with microbiomes, whereas we transferred only bacteria and viruses between microbiome generations to rule out any confounding effects of copropagated eukaryotes.

**Salt stress treatments and experimental contrasts.** Using different selection lines, we selected for beneficial microbiomes conferring salt tolerance to either sodium sulfate, $Na_2SO_4$, or aluminum sulfate,

$Al_2(SO_4)_3$. Such an experimental contrast of two treatments (here, two salt stresses) enables an experimenter to (i) compare evolving microbiomes using metagenomic time-series analyses, (ii) identify candidate microbes (indicator taxa) that differ between salt treatments and that may therefore confer salt tolerance to plants, and (iii) test the specificity of beneficial effects of evolved microbiomes in a cross-fostering experiment (described below).

**Control treatments.** To evaluate the effects of selection treatments, we included two nonselection control treatments. In the null control, we did not inoculate germinating seeds with any microbiomes, but microbes could enter soil from air, as was also the case for all other treatments. In the fallow-soil microbiome propagation control, we harvested microbiomes from fallow soil (no plant growing in a pot; microbiomes were harvested from root-free soil) and then propagated the harvested microbiomes to a pot with sterile fallow soil of the next microbiome generation. Specifically, each microbiome harvested from fallow soil was split, one part was propagated to sterile fallow soil to start the next microbiome generation, and another part of the same microbiome was applied to seeds planted in sterile soil to test the effect of such fallow-soil microbiomes on the growth of plants (details are in Text S1). Fallow-soil control is a nonselection treatment because a microbiome is transferred from exactly one pot in the previous generation to one pot in the next generation, resulting in enrichment (49) of microbes that proliferate under the specific salt conditions in soil but in the absence of higher-level microbiome selection that, in the selection treatment, selectively perpetuate growth-promoting microbiomes while discarding inferior microbiomes (i.e., there is no such discarding of inferior microbiomes in the fallow-soil control treatment).

**Number of selection cycles.** Our complete experiment involved one baseline generation (generation 0; Table S1) to establish initial microbiomes in replicate pots; eight rounds of differential microbiome propagation (generations 1 to 8; Table S1); and one final round (generation 9; Table S2) to evaluate the effects of the artificially selected microbiomes on seed set, for a total of 10 microbiome generations.

**Ramping of salt stress.** We increased salt stresses gradually during the selection experiment by (i) increasing between generations the molarity of the water used to hydrate dry soil before soil sterilization and planting (Text S1) and (ii) increasing correspondingly the molarity of the water that was added regularly to pots of growing plants to keep soils hydrated (Text S1). Over the 10 generations, sodium sulfate molarity in sodium stress treatments increased from 20 mM to 60 mM, and aluminum sulfate molarity in aluminum stress treatments increased from 0.02 mM to 1.5 mM (Text S1). The salt stresses of the baseline generation were chosen because, in pilot experiments, these stresses caused minimal delays in germination and growth compared to unstressed plants (Text S1). We did not preplan any maximum salt stresses that we wanted to reach via ramping within the 10 generations of microbiome propagation, because the salt stresses were increased judiciously each generation such that the plants would not be overstressed (because then beneficial microbiomes would not be able to ameliorate severe salt stresses) or understressed (and plants would then not need the help of beneficial microbiomes). The logic of increasing salt stresses stepwise between generations and decreasing salt stresses once between generations 5 and 6 when plants seemed overstressed (Fig. 2) is explained in the Text S1 under the subheading Soil Hydration and Salt Stress Treatments.

**Diversity of starter microbiome for baseline generation 0.** We prepared a single, well-mixed bacterial microbiome batch to inoculate all pots of the initial baseline generation 0, combining bacterial microbiomes from several rhizosphere sources to maximize the bacterial diversity of this starter inoculum. We used 2-$\mu$m Whatman filters to filter bacterial communities from root systems of three local grass species (*Bromus* sp., *Andropogon* sp., and *Eragrostis* sp.) and from root-systems of *B. distachyon* Bd3-1 plants used in earlier experiments (Text S1). We combined microbiomes from several sources in the hope of capturing a great diversity of bacteria, and we included microbiomes harvested from Bd3-1 roots to capture bacterial taxa that may be readily recruited by *B. distachyon* into its rhizosphere. This diverse starter microbiome changed during generation 0 through the aforementioned ecological processes once associated with a plant. The resulting variation in microbiomes between experimental replicates contributed to the variation in plant growth that we used for indirect selection on microbiome properties.

**Statistical analyses: plant biomass, generations 1 to 8.** We performed all analyses in R v3.3.1. We assessed differences in aboveground plant biomass (dry weight) among treatments of generations 1 to 8 by fitting the data to a generalized linear mixed model with a gamma error distribution. Statistical significance in the generalized linear mixed models was assessed with likelihood ratio tests and Tukey tests employed for posthost comparisons of treatment means (more details are in Text S2).

**Statistical analyses: total seed weight, generation 9.** Because plants were severely salt stressed in generation 9 and many plants therefore did not flower or produced very few seeds, the distribution of data was not normal (Fig. S5, top left). We attempted several data transformations to achieve approximate normality, but none of these transformations generated a distribution that approximated normality (Fig. S5b to d). We therefore used Kruskal-Wallis tests for nonparametric evaluation of differences between treatments in generation 9, and we used Mann-Whitney U tests for nonparametric *post hoc* comparisons between treatment means, correcting *P* values using the false discovery rate. All tests were two-tailed with alpha of 0.05 (more details are in Text S2).

**Data availability.** All data are available in Tables S1 and S2. All methods are described in detail in Text S1.

## SUPPLEMENTAL MATERIAL

Supplemental material is available online only.

**TEXT S1**, PDF file, 0.7 MB.
**TEXT S2**, PDF file, 0.2 MB.
**TEXT S3**, PDF file, 0.1 MB.

**TABLE S1**, PDF file, 0.7 MB.
**TABLE S2**, PDF file, 0.1 MB.

## ACKNOWLEDGMENTS

We thank Shane Merrell for help with greenhouse maintenance, Michael Mahometa for statistical advice, John Willis for suggesting the fallow-soil control treatment, and Hannah Marti for suggesting the thermostat metaphor. For constructive comments on the manuscript, we are grateful to John Vogel, Joey Knelman, Scott Carlew, Hannah Marti, Rong Ma, Emma Dietrich; the Juenger Lab; two anonymous reviewers; and participants of several Microbiome Selection workshops.

The work was supported by the National Science Foundation (awards DEB1354666 and DEB1911443 to U.G.M.), the Undergraduate Fellowship Program of the University of Texas Austin (to K.B.), the U.S. Department of Agriculture (award NIFA-2011-67012-30663 to D.L.D.), and the Stengl Endowment of the University of Texas Austin. The microbiome-selection methods were developed by U.G.M. in 2012 when he was a Fellow of the Japan Society for the Promotion of Science (award number 11186) visiting the Okinawa Institute of Science & Technology, hosted by JSPS-hosts Sasha Mikheyev and Kazuki Tsuji.

U.G.M., D.L.D., K.B., and T.E.J. developed the plant methods; U.G.M. developed the microbial and microbiome selection methods; T.E.J. contributed equipment and Bd3-1 seeds; U.G.M., M.R.K., and A.L.C. conducted experiments; A.L.C. and M.R.K. recorded all data (dry weights) blindly; J.A.E., C.C.S., and C.C.F. analyzed the data and designed figures; U.G.M. led the writing of the manuscript.

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
