## [Reviewer comments · mSystems]

Artificial Selection on Microbiomes to Breed Microbiomes That Confer Salt-Tolerance to Plants

Ulrich Mueller, Thomas Juenger, Melissa Kardish, Alexis Carlson, Kathleen Burns, Joseph Edwards, Chad Smith, Chi-Chun Fang, and David Des Marais

Corresponding Author(s): Ulrich Mueller, University of Texas at Austin

Review Timeline:

Submission Date:	September 10, 2021
Editorial Decision:	October 1, 2021
Revision Received:	October 20, 2021
Accepted:	October 22, 2021

Editor: Ashley Shade

Reviewer(s): The reviewers have opted to remain anonymous.

Transaction Report:

DOI: <https://doi.org/10.1128/mSystems.01125-21>

October 1, 2021

Prof. Ulrich G. Mueller
University of Texas at Austin
Integrative Biology
2401 Speedway #C0930
Austin, TX 78712

Re: mSystems01125-21 (Artificial Selection on Microbiomes to Breed Microbiomes That Confer Salt-Tolerance to Plants)

Dear Prof. Ulrich G. Mueller:

Thank you for submitting your manuscript to mSystems. We have completed our review and I am pleased to inform you that, in principle, we expect to accept it for publication in mSystems. However, acceptance will not be final until you have adequately addressed the reviewer comments.

My opinion is that the mentioned paired piece in preparation describing the microbiome data would also be a good fit for mSystems and, if you do decide to submit here, please include me in your handling editor request.

Please address the very minor comments of Reviewer 2. Thank you!

Preparing Revision Guidelines

Sincerely,

Ashley Shade

Editor, mSystems

Journals Department
Reviewer comments:

Reviewer #1 (Comments for the Author):

Nothing needs to be changed.

Reviewer #2 (Comments for the Author):

The authors demonstrate the power of microbiome engineering using the grass *Brachypodium distachyon* under two different salt treatments (sodium and aluminum). I find their methodology intriguing that addresses a large gap in the literature. Their experiment is composed of an impressive amount of work with striking results.

Line comments

Ln 68: distinguish Panke-Buisse as one of two studies

Ln 74 - 80: change semicolons to numbers. These points seem to be related to lns 271-345. Can they be organized in such a way to make the connection?

Ln 86 what processes?

Ln 94-98: Can you provide references for priority effects and increasing host-control? I believe they are in the M&M (lns 363-365)

Ln 102 awkward placement of in-text citations. Better fit ln 104 after clause

Ln 399: please provide filter sizes to encourage transparency and reproducibility

Ln 418-421 Can you please describe how fallow control biomass was measured here. I see it buried in supplemental, but this is a key point to your findings. My initial thought was 'How is relative plant mass calculated against the fallow control if the fallow control has no plants?'

Ln 433: 'also the molarity' 'allows the molarity'?

Ln 435-436: what were the intervals of increase? Why is molarity decreasing from gen 5 to 6 (shown in figure 2)

Ln 437: What is the justification for the top end of molarity?

Can you please add a brief version of the data analysis section in main text?

Figure 1:

It seems odd to have the exact same images in Figure 1 and S1.

Figure 2

The difference between a&c and b&d could be emphasized better. Based on the axes, they are depicting the same thing. Can a legend be added to distinguish the two colors? Better yet, could a small image be added for fallow vs. null controls?

Response to Reviewers

Reviewer comments in black

Our responses below in blue

All changes made to the reviewed manuscript are highlighted in TrackChanges in a PDF/word.doc submitted with the revised finalized manuscript.

Reviewer #1 (Comments for the Author):

Nothing needs to be changed.

Reviewer #2 (Comments for the Author):

The authors demonstrate the power of microbiome engineering using the grass *Brachypodium distachyon* under two different salt treatments (sodium and aluminum). I find their methodology intriguing that addresses a large gap in the literature. Their experiment is composed of an impressive amount of work with striking results.

Line comments

Ln 68: distinguish Panke-Buisse as one of two studies

Our response: Good idea, and thank you for this suggestion. We have split the three references that appeared together in the preceding sentence, and now split these into the two references for the two studies on *Arabidopsis* (to appear now in the next sentence on *Arabidopsis*) and the single reference for the one study on wheat (to appear in the subsequent sentence on wheat).

Ln 74 - 80: change semicolons to numbers. These points seem to be related to lines 271-345. Can they be organized in such a way to make the connection?

Our response: The additional experiments that we outline for future research in lines 271-345 of our Discussion go well beyond the new methods that we introduce in our study here (there is no one-to-one correspondence between these two sections). We therefore decided not to change the semicolons to numbers.

Ln 86 what processes?

Our response: We agree and we changed the wording to “the process of microbiome selection”

Ln 94-98: Can you provide references for priority effects and increasing host-control? I believe they are in the M&M (lines 363-365)

Our response: Good suggestion. We cite now several references on priority effects in microbiome assembly (Goldford et al 2018; Coyte et al 2015, 2021; Estrela et al 2021) and two references on host control (Sachs et al 2004; Coyte et al 2021).

Ln 102 awkward placement of in-text citations. Better fit Ln 104 after clause

Our response: We retained the in-text citation at the end of the subclause, because citing these references at the end of this clause (i.e., at the end of the full sentence) would then incorrectly suggest that these references discussed the host as a kind of “thermostat” to help gauge and adjust the “temperature” of its microbiomes (while in fact, our discussion here is the first, to our knowledge, to suggest the analogy of the host as a kind of “thermostat”).

Ln 399: please provide filter sizes to encourage transparency and reproducibility

Our response: Filter sizes are mentioned in the Supplemental Material, but we agree that it will help readers if filter sizes are also mentioned in the main article. We have added this information (2µm Whatman filters) at two places in the Methods of the main article.

Ln 418-421 Can you please describe how fallow control biomass was measured here. I see it buried in supplemental, but this is a key point to your findings. My initial thought was 'How is relative plant mass calculated against the fallow control if the fallow control has no plants?'

Our response: We agree. We have added the following sentence to the Methods in the main text: “Specifically, each microbiome harvested from fallow soil was split, one part was propagated to sterile fallow soil to start the next microbiome generation, and another part of the same microbiome was applied to seeds planted in sterile soil, to test the effect of such fallow-soil microbiomes on the growth of plants; details in Supplemental Material.”

Ln 433: 'also the molarity' 'allows the molarity'?

Our response: We now write “increasing correspondingly also the molarity of the water that was added regularly to pots and growing plants to keep soil hydrated”.

Ln435-436: what were the intervals of increase? Why is molarity decreasing from gen 5-6 (shown in figure 2) Ln437: What is the justification for the top end of molarity?

Our response: The changes in salt stress/molarity are explained in great detail on page 5 in the Supplemental Material under subheading Soil Hydration & Salt-Stress Treatments. We added therefore the following statement to the main text that refers to this subsection: “We did not pre-plan any maximum salt stresses that we wanted to reach during ramping within the 10 generations of microbiome propagation, because the salt stresses were increased judiciously each generation such that the plants would not be overstressed (because beneficial microbiomes would not be able to ameliorate severe salt stresses) nor plants would be understressed (and plants would then not need the help of beneficial microbiomes). The logic of increasing salt stresses stepwise between generations, and once decreasing salt stresses between Generations 5 & 6 when plants seemed overstressed (Figure 2), is explained in the Supplemental Material under the subheading Soil Hydration & Salt-Stress Treatments.”

Can you please add a brief version of the data analysis section in main text?

Our response: We added a section under subheading “Statistical analyses” towards the end of the Methods section.

Figure 1: It seems odd to have the exact same images in Figure 1 and S1.

Our response: We wrote the Supplemental Material document so it could stand on its own (as this document will be available online as a separate file), so readers would not need to refer back to the main article to look up a figure to understand details. We therefore decided to duplicate in the Supplementary Material the key figure visualizing the microbiome-selection cycle.

Figure 2: The difference between a&c and b&d could be emphasized better. Based on the axes, they are depicting the same thing. Can a legend be added to distinguish the two colors? Better yet, could a small image be added for fallow vs. null controls?

Our response: The difference between a&c (left column) and b&d (right column) is already indicated by the labels at the top, “Sodium-Salt Tolerance” (left column) and “Aluminum-Salt Tolerance” (right column). Values from fallow and null controls were used to relativize the measurements from the selection treatments (plants in fallow and null controls perform at the levels of the dashed horizontal lines, as explained in the figure caption), so it does not make sense to add images for fallow and null controls. To increase clarity of the difference between a&c versus b&d, we added the following sentence at the beginning of the caption of Figure 2: “Microbiomes were selected in two concurrent experiments under sodium-salt stress (left column) and aluminum-salt stress (right column)”.

October 22, 2021

Prof. Ulrich G. Mueller
University of Texas at Austin
Integrative Biology
2401 Speedway #C0930
Austin, TX 78712

Re: mSystems01125-21R1 (Artificial Selection on Microbiomes to Breed Microbiomes That Confer Salt-Tolerance to Plants)

Dear Prof. Ulrich G. Mueller:

Your manuscript has been accepted, and I am forwarding it to the ASM Journals Department for publication. For your reference, ASM Journals' address is given below. Before it can be scheduled for publication, your manuscript will be checked by the mSystems senior production editor, Ellie Ghatineh, to make sure that all elements meet the technical requirements for publication. She will contact you if anything needs to be revised before copyediting and production can begin. Otherwise, you will be notified when your proofs are ready to be viewed.

As an open-access publication, mSystems receives no financial support from paid subscriptions and depends on authors' prompt payment of publication fees as soon as their articles are accepted. =

Publication Fees:

We recognize that the video files can become quite large, and so to avoid quality loss ASM suggests sending the video file via <https://www.wetransfer.com/>. When you have a final version of the video and the still ready to share, please send it to Ellie Ghatineh at eghatineh@asmusa.org.

Sincerely,

Ashley Shade
Editor, mSystems

Journals Department
Supp File 3 Supplemental Material Results: Accept

SuppFile 5 Supplemental Table S2: Accept

SuppFile 2 Supplemental Material Statistical Analyses: Accept

SuppFile 4 Supplemental Table S1: Accept

SuppFile 1 Supplemental Material Methods: Accept